# Epitope Variation in Hemagglutinin and Antibody Responses to Successive A/Victoria A(H1N1) Strains in Young and Older Adults Following Seasonal Influenza Vaccination: A Pilot Study

**DOI:** 10.3390/vaccines13070774

**Published:** 2025-07-21

**Authors:** Mónica Espinar-García, Isabel María Vallejo-Bermúdez, María Ángeles Onieva-García, Irene Reina-Alfonso, Luis Llapa-Chino, Pablo Álvarez-Heredia, Inmaculada Salcedo, Rafael Solana, Alejandra Pera, Alexander Batista-Duharte

**Affiliations:** 1Department of Cell Biology, Physiology and Immunology, University of Cordoba, Av. Menendez Pidal s/n, 14004 Cordoba, Spain; b32esgam@uco.es (M.E.-G.); b82vabei@uco.es (I.M.V.-B.); z02reale@uco.es (I.R.-A.); z42llchl@uco.es (L.L.-C.); rsolana@uco.es (R.S.); 2Immunology and Allergy Group (GC01), Maimonides Biomedical Research Institute of Cordoba (IMIBIC), University of Cordoba, Reina Sofia University Hospital, Av. Menendez Pidal s/n, 14004 Cordoba, Spain; pablo.alvarez@imibic.org; 3Preventive Medicine and Public Health Unit, Maimonides Biomedical Research Institute of Cordoba (IMIBIC), University of Cordoba, Reina Sofia University Hospital, Av. Menendez Pidal s/n, 14004 Cordoba, Spain; mariaa.onieva.sspa@juntadeandalucia.es (M.Á.O.-G.); minmaculada.salcedo.sspa@juntadeandalucia.es (I.S.); 4Department of Medical and Surgical Sciences, University of Cordoba, Av. Menendez Pidal s/n, 14004 Cordoba, Spain

**Keywords:** influenza vaccine, original antigenic sin (OAS), antigenic imprinting, B-cell epitope mapping, hemagglutinin (HA), immune response, antibody response, cytomegalovirus (CMV)

## Abstract

**Background:** Annual influenza vaccine updates target viral drift, but immune responses may be biased by original antigenic sin (OAS). Few studies have explored this across closely related strains. This study examines how OAS shapes responses to sequential influenza variants in the context of seasonal vaccination. **Methods:** We conducted a prospective, longitudinal study to assess the humoral immune response to the 2023–2024 seasonal influenza vaccine containing the A/Victoria/4897/2022 (H1N1) strain. Bioinformatic analyses compared the hemagglutinin (HA) sequences of A/Victoria/4897/2022 and the antigenically related A/Victoria/2570/2019 strain. B-cell epitopes were mapped with BepiPred-3.0 and BepiBlast, and their physicochemical properties analyzed via accessibility, β-turns, flexibility, and hydrophilicity. Antibody responses were measured pre- and 28 days post-Vaxigrip Tetra vaccination in young (18–35) and older (>65) adults, stratified by cytomegalovirus (CMV) serostatus. HA sequences showed >97% identity, with variations mainly in the globular head. Predicted B-cell epitopes overlapped variable sites, suggesting possible immune escape. Despite having been vaccinated against the 2022 strain, serology showed higher antibody titers against the 2019 HA strain in all participants. This pattern suggests a potential antigen imprinting effect, though confirmation awaits further analysis. Age groups differed: older adults showed greater variability, while younger CMV+ individuals tended toward stronger 2019 HA responses. **Conclusions:** These findings suggest a complex interplay of factors shaping immune responses, though the imprinting effect and the potential role of CMV warrant further exploration in larger, more focused studies.

## 1. Introduction

Influenza infection in human populations is characterized by seasonal epidemics occurring in the winter and very few cases observed during the summer in temperate regions of both hemispheres and year-round endemicity with unpredictable variation in intensity in tropical and subtropical zones [1,2]. Sustained transmission of Influenza A viruses (IAVs) within the human population exerts selective pressure that promotes the emergence of immune escape variants [3]. This evolutionary dynamic is facilitated by the high genetic plasticity of IAVs, which is driven by elevated mutation rates, estimated at over 1 × 10^−3^ substitutions per nucleotide site per year [4]. Consequently, amino acid substitutions in the antigenic regions of the surface glycoproteins hemagglutinin (HA) and neuraminidase (NA) may confer a selective advantage, allowing viral variants to evade recognition by pre-existing host immunity. This is especially important in human-adapted strains, where intense immune pressure drives viral evolution. Neutralizing antibodies directed against HA are crucial in preventing receptor binding and reinfection with antigenically similar strains. Over time, the progressive accumulation of such immune-evasive mutations in HA leads to a process known as antigenic drift, which plays a key role in changing the virulence of IAVs [5,6]. This process involves small, cumulative mutations in the HA gene that allow the virus to escape recognition by antibodies generated from prior infections or vaccinations. Thus, surveillance during epidemic periods aims to detect emerging strains capable of evading immune protection, indicating when vaccine updates may be necessary [7]. As a result, seasonal influenza vaccines are updated annually to keep up with the rapid evolution of influenza viruses, mainly driven by antigenic drift in the HA surface protein.

Although antigenic drift in IAVs has been recognized for almost eight decades [8] the specific contribution of different selective pressures to this process remains unclear [9]. In this context, the role of seasonal flu vaccination is under investigation, with evidence suggesting that vaccine-induced immune pressure in humans may favor the selection of influenza variants that are genetically distant from the vaccine strains [10].

One factor that shapes the immune response to influenza is antigenic imprinting—also known as original antigenic sin (OAS)—where the immune system preferentially activates memory responses to previously encountered strains, potentially limiting the effectiveness of responses to newer, updated vaccine strains. In the context of the influenza virus, OAS was first described in 1960 by Sir Thomas Francis, who established that the immune system’s response to a pathogen or antigen is heavily influenced by the first encounter with that pathogen or a related one [11]. In this way, the initial exposure to the influenza virus—typically occurring in early childhood—shapes how our immune system responds to future exposures, whether through infection or vaccination. OAS received limited scientific attention after its initial discovery, even during the 1968 H3N2 pandemic. Hoskins and colleagues, through studies conducted between 1970 and 1976, observed that individuals repeatedly vaccinated against influenza showed reduced protection against new strains compared to first-time vaccine recipients [12,13,14]. This unexpected finding, later termed the Hoskins paradox, raised concerns about the long-term effectiveness of annual influenza vaccination in the face of antigenic drift. In 2012, Lessler et al. reframed the concept of original antigenic sin as “antigenic seniority”, describing the stronger immune responses to influenza strains encountered early in life, not just the first exposure [15]. This terminology underscores the role of age in shaping immune responses to emerging variants. Later, the concept of antigenic imprinting was introduced to explain how early exposure to influenza hemagglutinin influences future responses to related strains. Notably, Gostic et al. (2016) demonstrated that childhood imprinting confers lasting protection against severe disease from novel avian influenza subtypes such as H5N1 and H7N9 [16]. This phenomenon is now understood within the broader framework of antigenic imprinting, encompassing both its protective and limiting effects on lifelong humoral immunity [17].

Despite growing recognition of the importance of antigenic imprinting in shaping immune responses to influenza, most studies have focused on immune responses to genetically distant viral variants. While these investigations have provided valuable insights, the immune response to those closely related variants differing by only a few antigenic mutations has not been sufficiently explored. Importantly, even minor amino acid changes in the hemagglutinin protein can interact in a non-additive manner, a phenomenon known as epistasis, which can potentially alter antigenicity in unpredictable ways [18]. These subtle yet consequential changes may significantly affect the quality and specificity of the humoral response, particularly in individuals with pre-existing immunity.

Cytomegalovirus (CMV) infection is highly prevalent and has a profound impact on the immune system, particularly in older individuals. CMV seropositivity has been associated with immune remodeling, including expansion of late-differentiated T cells and chronic low-grade inflammation [19,20]. These changes are commonly associated with immunosenescence and may affect the quality and magnitude of vaccine-induced responses. However, the impact of CMV on humoral immunity following influenza vaccination remains unclear, with some studies suggesting reduced efficacy [21,22,23], while others report no detrimental effect [24] or even enhanced responses [25]. Due to the lack of consensus on the role of latent CMV infection in influenza vaccine responses, considering CMV serostatus in future analyses may help clarify inter-individual differences in immunogenicity across age groups [24].

Considering the complexity of immune responses to antigenically drifted influenza strains, along with the impact of factors such as age, pre-existing immunity, and CMV serostatus, a pilot study represents a meaningful initial approach to investigate these variables in a controlled and hypothesis-generating context. By examining two closely related H1N1 strains from recent influenza vaccines (A/Victoria/4897/2022 and A/Victoria/2570/2019), we evaluate the impact of subtle HA mutations on antibody specificity and their potential to modulate antigenic imprinting. Stratifying responses by age and CMV serostatus offers initial insights into how immune history and aging affect vaccine-induced humoral responses. These findings will guide the design of larger studies to optimize vaccine composition and predict responses to emerging strains.

## 2. Materials and Methods

Influenza virus sequence data extraction and multiple sequence alignment were carried out. For bioinformatics analysis, the full-length amino acid sequences of the influenza A virus from the strains A/Victoria/2570/2019 (accession no. WEY08940.1) and A/Victoria/4897/2022 (accession no. WEY08903.1) were obtained from GenBank (http://ncbi.nlm.nih.gov/) and a Pairwise Sequence Alignment (PSA) was performed by EMBOSS Needle, available on the Job Dispatcher platform of the European Bioinformatics Institute (https://www.ebi.ac.uk/jdispatcher/psa) (accessed on 11 February 2025).

### 2.1. B-Cell Epitope Prediction

Linear B-cell epitopes were predicted using two tools: Linear B-cell epitopes were predicted using BepiPred-3.0 and BepiBlast. BepiPred-3.0 (https://services.healthtech.dtu.dk/service.php?BepiPred-3.0) (accessed on 12 February 2025) is a sequence-based tool that leverages numerical representations from the ESM-2 protein language model to enhance the prediction of linear and conformational B-cell epitopes significantly. By optimizing the model architecture, training strategy, and input features, and adopting an epitope annotation method from previous work, it achieves unprecedented predictive performance [26]. BepiBlast is an improved BLAST-based tool for B cell epitope prediction that enhances conventional sequence alignment by integrating epitope-annotated databases to identify homologous regions with potential antigenicity (http://imath.med.ucm.es/bepiblast/), last updated on 16 September 2022 (accessed on 14 February 2025), [27]. The Kolaskar and Tongaonkar Antigenicity tool was also used to predict antigenicity [threshold  =  1.016] [28].

### 2.2. Surface Accessibility and β-Turn Prediction

To evaluate the surface accessibility of the predicted epitopes, the Emini Surface Accessibility Prediction tool from the IEDB suite was employed [threshold =  1.000] [29]. Chou and Fasman beta-turn prediction was performed using a threshold =  1.007 to identify beta-turn regions within both proteins [30], as these structural motifs play a key role in enhancing antigenicity [31].

### 2.3. Evaluation of Flexibility and Hydrophilicity of B Cell Epitopes

Hydrophilicity and structural flexibility are known to correlate with the antigenic potential of peptides [32]. To evaluate these properties, the conserved B cell epitopes were analyzed using the Karplus and Schulz flexibility prediction tool [threshold =  1.000] [33] and the Parker hydrophilicity scale [threshold  =  1.678] [34], which assess peptide flexibility and hydrophilicity, respectively. The bioinformatics analysis procedure is outlined in the flowchart presented in Figure 1.

### 2.4. Study Design

A prospective, longitudinal study was conducted to perform an immunological analysis before and after the administration of the influenza vaccine Vaxigrip Tetra (Sanofi) during the 2023–2024 vaccination campaign. The study included 17 asymptomatic participants of both sexes, with no history of chronic diseases and/or immunosuppressive treatment. Participants were stratified by age group (young: 18–35 years; older: >65 years) and CMV serostatus (positive vs. negative). All subjects received the seasonal influenza vaccine containing the A/Victoria/4897/2022 (H1N1) hemagglutinin (HA) antigen.

All participants received a summary of the project and provided written informed consent before enrollment. The study was approved by the Provincial Research Ethics Committee of Córdoba, as stated in Act No. 340 dated 28 September 2022.

#### Sample Collection

Peripheral blood samples were collected from all subjects before and 30 days post-vaccination, following institutional ethical guidelines. Serum samples were stored at −80 °C until use.

Recombinant hemagglutinin (HA) proteins from the influenza virus strains A/Victoria/2570/2019 (H1N1)pdm09-like and A/Victoria/4897/2022 (H1N1)pdm09-like were obtained from MyBioSource (San Diego, CA, USA) (Ref: MBS434303, Lot: AT702EY) and Sino Biological (Beijing, China) (Ref: 40938-V08H, Lot: LC17OC0923), respectively. Both HA proteins were expressed in HEK293 cells, purified to >95% purity, and prepared under comparable conditions following the respective manufacturers’ protocols.

### 2.5. ELISA Procedure

Ninety-six-well ELISA plates (Nunc MaxiSorp, Thermo Fisher Scientific, Waltham, MA, USA) were coated overnight at 4 °C with 100 μL per well of HA protein diluted in PBS at a concentration of 1 μg/mL. Plates were then washed three times with PBS containing 0.05% Tween-20 (PBST) and blocked with 200 μL per well of 5% non-fat milk in PBST for 1 h at room temperature. After blocking, serum samples were serially diluted in PBST with 1% milk and added to the plates (100 μL per well). Plates were incubated for 2 h at room temperature, followed by three washes with PBST. Horseradish peroxidase (HRP)-conjugated rabbit anti-human IgG secondary antibody, diluted in PBST with 1% milk was added (100 μL per well) and incubated for 1 h at room temperature. After final washes, 100 μL of TMB substrate (3,3′,5,5′-Tetramethylbenzidine) was added to each well and the reaction was stopped after 10 min with 50 μL of 2 M H_2_SO_4_. Absorbance was measured at 450 nm using a microplate reader. Absorbance values were plotted against the log_10_ of the serum dilution to generate a binding curve. The area under the curve (AUC) was calculated using GraphPad Prism software, employing the trapezoidal method to quantify antibody binding responses. Background absorbance from blank wells was subtracted before analysis. All samples were run in duplicate and mean values were used for analysis.

To calculate the total IgG antibody titers of the serum samples, the mean optical density (OD) of the cell control wells, and their standard deviation (SD) were determined. The cut-off for the presence of influenza-specific IgG antibodies was then calculated using the following formula, where f represents a standard deviation multiplier based on the number of control samples and the selected confidence level [35].Cutoff = mean OD + (SD × f)

Here, the number of control samples was eight and the confidence level was 99.9%, which determines that f = 5.076 [36]. The endpoint titer for each sample was defined as the reciprocal of the highest dilution that yielded an absorbance above the cutoff.

### 2.6. CMV Serology

To assess prior exposure to CMV, a plasma or serum sample was analyzed using a chemiluminescent immunoassay (CLIA). The assay was performed on the Liaison^®^ XL automated platform (DiaSorin S.p.A., Saluggia, Italy), employing the Liaison^®^ CMV IgG II assay. This method quantitatively detects IgG antibodies specific to CMV, providing a sensitive and reliable measure of past CMV infection or immune status. This assay provides highly sensitive and reliable results, aiding in the diagnosis of past CMV infections and evaluating the immune response of the individual.

### 2.7. Statistical Analyses

Statistical analyses were conducted to compare antibody responses across participant groups and time points. A non-parametric approach was adopted for all statistical analyses. For paired comparisons (e.g., pre- vs. post-vaccination within the same individuals), the Wilcoxon signed-rank test was used. For unpaired comparisons between independent groups, a non-parametric version of the Student’s *t*-test—the Wilcoxon rank-sum test—was applied. All analyses were conducted using GraphPad Prism (Version 8.02, GraphPad Software, San Diego, CA, USA) and R software, version 4.3.3 (R Foundation for Statistical Computing, Vienna, Austria,. *p*-value ≤ 0.05 was considered statistically significant.

## 3. Results

### 3.1. Pairwise Sequence Alignment of Hemagglutinin (HA) from A/Victoria/2570/2019 (Accession No. WEY08940.1) and A/Victoria/4897/2022

Pairwise sequence analysis showed a high degree of conservation between the compared sequences, with 97.5% identity and 99.1% similarity, with no gaps. These results indicate that the analyzed proteins share a high degree of homology, not only in terms of exact residue matches but also in terms of the conservation of relevant physicochemical properties, suggesting strong evolutionary pressure to maintain their structure and function. The absence of insertions or deletions reinforces the hypothesis that these proteins have diverged minimally, consistent with a shared or closely related biological function (Figure 2).

Despite the overall high sequence similarity between the sequences analyzed, several amino acid substitutions were observed, primarily within the globular head domain of HA. This region, known to be the most antigenically variable portion of the protein, exhibited some differences that may influence antibody binding. 

### 3.2. Linear B-Cell Epitopes Mapping Across Variants

As shown in Figure 3, B-cell epitope prediction using the BepiPred-3.0 algorithm revealed multiple regions within the HA proteins of A/Victoria/2570/2019 and A/Victoria/4897/2022 with high immunogenic potential. These predicted epitopes, highlighted in yellow, were primarily located within the globular head domain and notably overlapped with sites of amino acid variation between the two strains. This suggests that even minor sequence differences may influence the epitope landscape and thereby modulate antibody recognition. To further characterize the nature and conservation of these predicted epitopes, we employed BepiBlast, a sequence alignment-based tool designed to map homologous B-cell epitopes across related antigens. The results, presented in Figure 4, provided additional resolution by identifying both conserved and unique epitope regions, thereby supporting the hypothesis that antigenic variation in these immunodominant regions may underlie the observed imprinting bias.

### 3.3. Surface Accessibility and β-Turn Prediction

Surface accessibility was assessed using the Emini Surface Accessibility Prediction method. Regions scoring above the threshold (1.000) were identified, indicating high surface exposure. Furthermore, β-turn prediction using the Chou and Fasman method (threshold 1.007) identified several flexible structural motifs, which are likely to enhance antigenicity (Figure 2).

### 3.4. Flexibility and Hydrophilicity Prediction

Structural flexibility of the epitopes was evaluated using the Karplus and Schulz method, identifying regions with scores above the 1.000 threshold, suggesting conformational flexibility favorable for antigen–antibody interaction. Hydrophilicity analysis using the Parker scale revealed hydrophilic regions (scores > 1.678), further supporting the immunogenic potential of the selected sequences (Figure 5).

### 3.5. Serological Analysis

Both young and older individuals exhibited significantly higher antibody titers against the 2019 HA antigen compared to the 2022 HA antigen, despite the latter being the actual vaccine strain administered during the study period. Quantitative serological data revealed that the 2019/2022 titer ratio was consistently above one across all participants, with no individual showing a dominant response to the updated antigen (Figure 6). These findings may reflect immune imprinting, where prior exposure to antigenically similar strains enhances the memory B cell response to conserved epitopes, potentially limiting the response to new epitopes in the current vaccine. However, as this is a pilot study, these findings require confirmation through protein quality analyses. Such analyses are essential to assess whether structural or conformational variations, independent of the primary sequence, contributed to the observed immune response patterns. 

To comparatively analyze the antibody responses across age groups, we assessed antigen-specific IgG levels by ELISA at 28 days post-vaccination and calculated the area under the curve (AUC) from serial dilutions to capture both titer magnitude and antibody-binding strength. All individuals mounted a measurable and specific response following vaccination. Although older adults tended to exhibit lower antibody AUC values compared to younger participants, this difference did not reach statistical significance. Notably, the older group displayed greater interindividual variability in their responses, suggesting underlying heterogeneity in immune competence with advancing age. This variability may reflect differences in immunological history, pre-existing immunity, comorbidities, or immune remodeling associated with aging (Figure 7).

To further investigate the potential modulatory role of CMV serostatus on antigen-specific responses, all participants were stratified into CMV-negative and CMV-positive subgroups. Among younger individuals, those who were CMV+ positive tended to exhibit higher antibody titers against the 2019 HA antigen. However, due to the reduced number of samples after stratification, especially in subgroup analyses, this trend should be interpreted with caution. No consistent pattern was observed among older individuals concerning CMV status.

To assess the humoral immune response induced by influenza vaccination, IgG titers specific to hemagglutinin (HA) from the 2019 and 2022 strains were measured in young and older adults. An additional analysis, shown in Appendix A, corroborated these findings: both age groups exhibited a significant increase in IgG titers following vaccination. However, baseline and post-vaccination titers were consistently lower in older adults compared to young individuals, particularly in response to the 2022 strain, suggesting an age-related decline in vaccine-induced antibody responses.

## 4. Discussion

Seasonal IAV evolution is primarily driven by antigenic drift, whereby the accumulation of mutations in surface proteins enables the emergence of antigenically distinct strains. As population-level immunity builds against circulating variants, these new strains gain a selective advantage, gradually replacing prior ones and necessitating frequent updates to the composition of seasonal influenza vaccines [17,37]. To escape immune detection, the virus accumulates mutations in the HA protein, particularly in surface-exposed regions, that prevent antibody binding while preserving viral function [38,39]. This process, known as antigenic drift, primarily affects the HA head, which binds to the sialic acid receptor. Every 2 to 10 years, these mutations can lead to substantial antigenic shifts in circulating IAVs, forming new antigenic “clusters” that reduce the protective effect of existing immunity. Notably, only a few amino acid changes near the HA receptor-binding site are often sufficient to drive these cluster transitions [5,6,38,40].

In this study, we explored whether antigenic imprinting could be detected between two closely related influenza A(H1N1) strains circulating in successive seasons, despite their high degree of sequence similarity, as a potential explanation for the differential antibody responses observed.

To establish the occurrence of B cell imprinting, several stringent criteria must be satisfied, as outlined by King et al. (2023): (1) clear evidence of prior immune exposure (e.g., antibodies or memory B cells specific to a previous antigen), (2) shared epitopes between the original and new antigens, (3) inclusion of modified regions in the updated antigen, and (4) a biased immune response toward conserved epitopes upon re-exposure [41].

The findings of this study suggest that immune imprinting may play a role in shaping antibody responses to influenza HA, although further research is needed to confirm and elucidate this effect. While the sequence differences between the HA proteins of A/Victoria/2570/2019 and A/Victoria/4897/2022 were minimal and primarily localized within the globular head domain, these subtle substitutions significantly influenced B cell epitope properties. This observation aligns with previous studies demonstrating that even single amino acid changes in surface-exposed residues can markedly alter antigenicity and immune recognition [5,38]. The globular head region of HA, which includes the receptor-binding site, is subject to intense immune selection pressure and frequently serves as a hotspot for antigenic drift due to its role as the main target of neutralizing antibodies [42,43].

To further characterize these changes, B-cell epitope prediction tools such as BepiPred-3.0 and BepiBlast identified that many of the mutations aligned with regions of high immunogenic potential. Complementary biophysical analyses, including the Emini [29], Parker [34], Karplus–Schulz [33], and Chou–Fasman [30] algorithms, revealed consistent alterations in surface accessibility, flexibility, and hydrophilicity between the two HA variants. These structural and biophysical properties are known to influence the likelihood of antigen–antibody interactions [44,45]. For instance, a study by Zhu et al. (2022) identified a key amino acid mutation, A180V, in the HA protein of the H9N2 avian influenza virus. This substitution, located at a non-conserved site within the receptor-binding domain, was directly responsible for viral antigenic variation. The A180V mutation enhanced receptor binding activity, leading to a significant decrease in cross-reactivity to immune sera from animals immunized with various H9N2 strains. Although this mutation did not physically prevent antibody binding, it promoted viral escape from neutralizing reactions and slightly affected in vivo cross-protection, suggesting its role in the adaptive evolution of the virus [46]. Similarly, Xu et al. (2022) identified three amino acid substitutions (at positions 190, 230, and 269) in the Eurasian avian-like H1N1 swine influenza virus (EA H1N1 SIV) A/swine/Henan/11/2005 (HeN11) that enabled the virus to evade neutralization by monoclonal antibodies and contributed to antigenic drift [47]. Another study by Wang et al. (2021) provides experimental evidence that the G158E mutation in the HA significantly affects the antigenic properties of Eurasian avian-like H1N1 swine influenza viruses (EA H1N1 SIVs). Structural analysis suggested that the substitution from glycine (G) to glutamic acid (E) at position 158 introduced steric hindrance and increased hydrophilicity on the HA protein surface, contributing to antigenic drift in EA H1N1 SIVs [48].

In our study, the serological analysis showed a consistent dominance of the antibody response toward the 2019 HA antigen across age groups, despite administration of the 2022 vaccine strain. Prior studies have demonstrated that antigenic similarity between historical and contemporary influenza strains can lead to preferential expansion of cross-reactive memory B cells, resulting in a “back-boosting” of responses to the priming antigen [16,49]. Such imprinted responses can potentially undermine the efficacy of updated vaccines, particularly when the immune system fails to adequately respond to novel epitopes introduced through antigenic drift [50]. However, the findings described here should be interpreted with caution, as this is a pilot study. Although a possible antigen imprinting effect was observed, these results are preliminary and require confirmation through protein quality analyses, including assessments of structural or affinity variations beyond those attributable to primary sequence changes caused by mutations.

Interestingly, while older adults exhibited greater interindividual variability in antibody titers, the average responses did not differ significantly from those of younger individuals. This may reflect the multifactorial nature of immunosenescence, which includes not only diminished antibody production but also alterations in B cell repertoire diversity and function [51,52]. However, the absence of statistically significant age-associated differences in this cohort could be due to sample size limitations or unaccounted variability in exposure history.

Another intriguing aspect of this study was the trend toward higher antibody titers in CMV-seropositive younger individuals. CMV has co-evolved with humans over thousands of years and now constitutes a significant component of the human virome, a complex ecosystem of both commensal and pathogenic viruses, much like the microbiome [53,54]. Following primary infection, CMV establishes lifelong latency and is known to drive chronic immune activation and reshaping of the immune landscape, particularly in older adults. CMV seropositivity has been associated with altered responses to vaccination; however, the nature, consistency, and clinical relevance of this impact remain unclear. A range of studies analyzed in a systematic review show no consistent evidence that latent CMV infection influences the antibody response to influenza vaccination. While some findings suggest an enhanced response, others indicate a reduced effect, and several report no impact at all [55]. Few studies have investigated the potential link between CMV infection and antigen imprinting [56,57]. Therefore, further research is needed to clarify the potential role of latent CMV infection in modulating vaccine-induced immunity

While originally recognized in influenza due to its frequent seasonal outbreaks and rapid mutation rate, the phenomenon of immune imprinting has also been identified in infections caused by other viruses, such as dengue [58] norovirus [59], hepatitis C virus (HCV) [60], respiratory syncytial virus (RSV) [61,62] and, more recently, SARS-CoV-2. In the context of SARS-CoV-2, several studies have documented imprinting effects from prior seasonal coronavirus exposure or earlier variants that shape responses to newer strains or updated vaccines [63,64,65]. Interestingly, early CMV seropositivity appears to enhance both cellular and humoral immune responses, potentially increasing the magnitude and complexity of immunological imprinting in younger individuals. This effect may be driven by heterologous immunity mechanisms, whereby CMV shapes immune responsiveness to unrelated antigens.

Several mechanisms could explain our findings. First, the chronic low-grade inflammation associated with CMV infection may function as an endogenous adjuvant, enhancing antigen presentation and immune activation upon vaccination [25,66]. CMV also promotes a sustained inflammatory milieu (characterized by elevated levels of IL-6, TNF-α, and C-reactive protein), which can modulate immune cell function and differentiation. While such an environment has been associated with impaired B cell responses in older adults, it may contribute differently in younger individuals, potentially facilitating stronger or more focused immune imprinting upon initial antigen exposures [25]. In this context, it is plausible that CMV seropositivity may enhance antigenic imprinting in younger individuals by shaping a more reactive or “primed” immune landscape during early encounters with influenza antigens. This hypothesis aligns with emerging evidence suggesting that CMV-driven immune activation can augment responses to heterologous pathogens.

In a previous study, it was observed that CMV+ young adults had higher levels of CD57+ T cells and exhibited more polyfunctional responses than CMV− individuals [63,64,65]. These findings support the idea that CMV-induced immune modulation could amplify imprinting effects, with important implications for vaccine efficacy in younger people. In this context, a stronger imprinting response may predispose individuals to suboptimal reactions to antigenically drifted strains, as observed with SARS-CoV-2 variants [63,64,65]. Thus, considering CMV status during early-life vaccination planning could be crucial for improving long-term protection and shaping more adaptable immune responses.

Our results reinforce the need to consider imprinting history when designing vaccines, particularly in efforts aimed at inducing broad or universal protection. Strategies targeting conserved domains, such as the HA stalk, have shown promise in generating cross-protective memory B cells capable of recognizing diverse influenza subtypes [67,68].

*Limitations of this study:* This pilot study aimed to explore how specific amino acid changes between two closely related influenza A(H1N1) variants influence the specificity of the antibody response to the full-length HA protein. We did not assess responses to individual antigenic regions or fragments, nor did we evaluate the protective capacity of the elicited antibodies. Additionally, the relatively small sample size, particularly after being stratified by CMV serostatus, limited the statistical power of subgroup analyses. It also remains unclear whether the observed differences in binding signals between HA variants reflect true immunological differences, such as those related to immune imprinting, or are instead influenced by variations in protein quality or folding, independent of sequence changes. Biophysical and structural validation of the recombinant HA proteins will be required to address this.

*Future directions*: The next research phase will begin with the validation of B-cell epitopes to determine whether the observed differences in antibody binding are due to immune imprinting or variations in protein quality. This will be followed by the study of T-cell epitopes, particularly CD8+ T-cells, to gain a broader understanding of both humoral and cellular immune responses. The analysis will focus on T-cell epitopes for relevant HLA-A, -B, and -C alleles to better understand the impact of prior exposures on immunity. Additionally, cellular immune responses will be examined in more detail, considering age groups and CMV serostatus. With a larger and more diverse sample, subgroup analyses will be strengthened, providing deeper insights into the modulatory effects of CMV and prior exposures. These studies will ultimately contribute to refining influenza vaccine strategies by considering both antibody and cellular immunity, as well as the role of antigenic imprinting.

## Figures and Tables

**Figure 1 vaccines-13-00774-f001:**
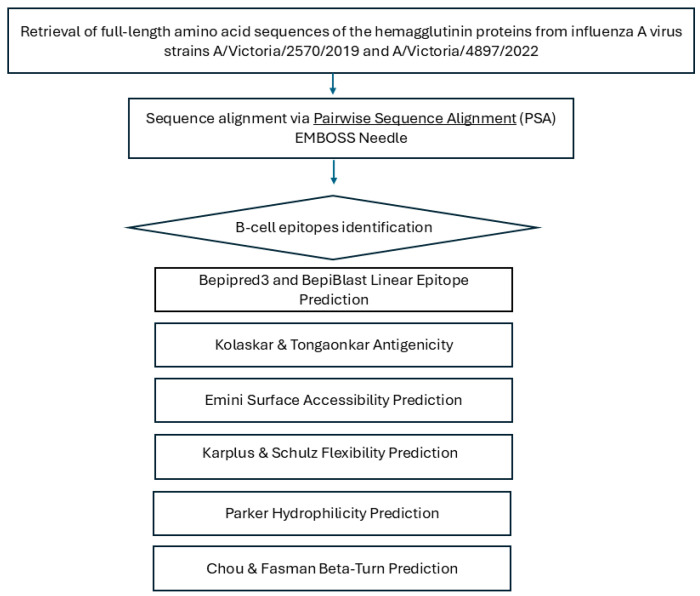
Schematic overview of the bioinformatics workflow used for B-cell epitope prediction in influenza A hemagglutinin. Full-length hemagglutinin (HA) amino acid sequences from strains A/Victoria/2570/2019 and A/Victoria/4897/2022 were retrieved from GenBank and aligned using the EMBOSS Needle pairwise sequence alignment tool. Predicted linear B-cell epitopes were identified with BepiPred-3.0, BepiBlast, and the Kolaskar–Tongaonkar method. Antigenic features—including surface accessibility, β-turns, flexibility, and hydrophilicity—were evaluated using Emini, Chou–Fasman, Karplus–Schulz, and Parker tools to assess immunogenic potential.

**Figure 2 vaccines-13-00774-f002:**
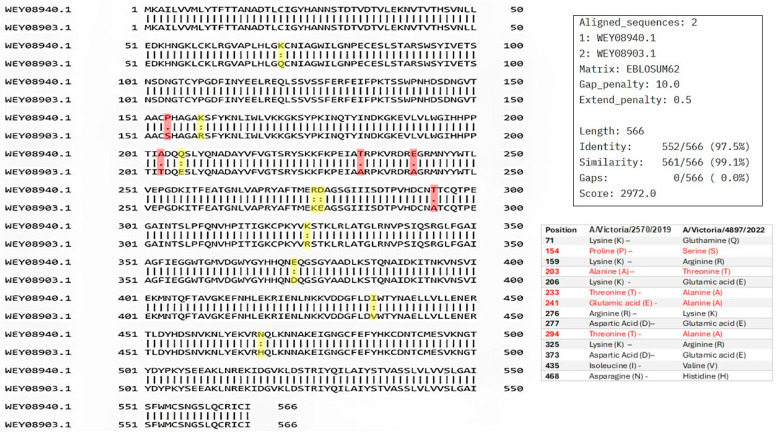
Pairwise sequence alignment of hemagglutinin (HA) from A/Victoria/2570/2019 (accession no. WEY08940.1) and A/Victoria/4897/2022 (accession no. WEY08903.1) strains using Pairwise Sequence Alignment (PSA) performed by EMBOSS Needle, available on the Job Dispatcher platform of the European Bioinformatics Institute. Mutations between the two sequences are highlighted. A colon (:) denotes strongly similar residues, while a period (.) indicates weakly similar residues. Highlighted positions are shown in yellow (strong similarity) and red (weak similarity).

**Figure 3 vaccines-13-00774-f003:**
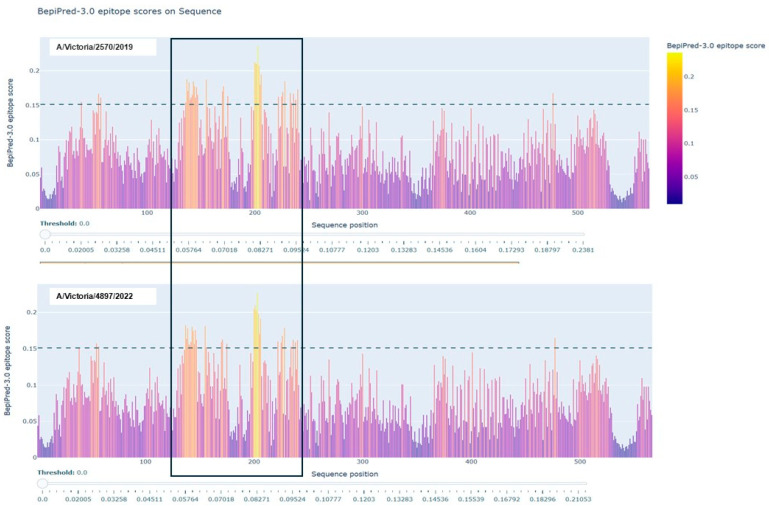
B-cell epitope prediction of the A/Victoria/2570/2019 and A/Victoria/4897/2022 HA proteins using BepiPred-3.0. The hemagglutinin (HA) sequences from both influenza strains were analyzed using the BepiPred-3.0 online prediction tool. Regions highlighted in yellow indicate residues with high prediction scores, suggesting probable locations of linear B-cell epitopes. Notably, several predicted epitopes overlapped with sites of amino acid substitution between the two variants, indicating potential alteration of immunogenicity.

**Figure 4 vaccines-13-00774-f004:**
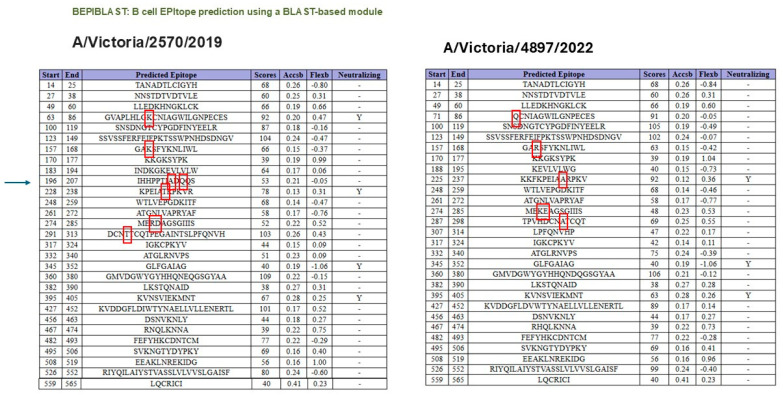
Epitope alignment analysis using BepiBlast for A/Victoria/2570/2019 and A/Victoria/4897/2022 HA proteins. BepiBlast was used to compare predicted B-cell epitopes between the two HA variants. The alignment highlights conserved and variable epitope regions. Red boxes indicate amino acid positions that differ between the two sequences, suggesting potential sites of altered immunogenicity relevant to antigenic drift and immune imprinting. The arrow in the left panel indicates that the changes occurring in the 196–207 segment led to the loss of the B-cell epitope property in that region, according to the bioinformatics program.

**Figure 5 vaccines-13-00774-f005:**
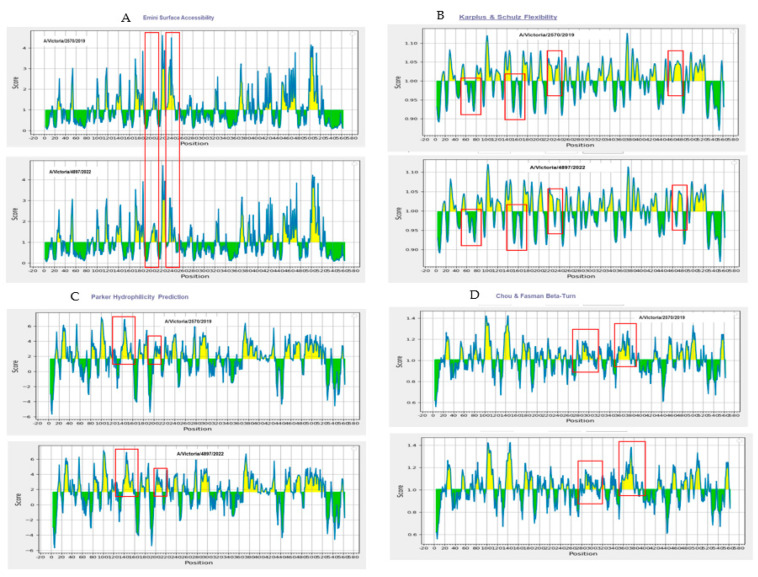
Biophysical characterization of predicted B-cell epitopes based on surface accessibility, β-turns, flexibility, and hydrophilicity. Surface accessibility was assessed using the Emini Surface Accessibility Prediction method, with regions scoring above the 1.000 threshold indicating high surface exposure (**A**). Epitope flexibility was evaluated using the Karplus and Schulz method, where scores above 1.000 denote conformational mobility favorable for antibody binding (**B**). Hydrophilicity, analyzed via the Parker scale, revealed several regions exceeding the 1.678 threshold, suggesting strong immunogenic potential (**C**). β-turns were predicted using the Chou and Fasman method, identifying flexible structural motifs above the threshold of 1.007, which are typically associated with increased antigenicity (**D**). Red boxes indicate the regions with differences between the A/Victoria/2570/2019 and A/Victoria/4897/2022 HA proteins.

**Figure 6 vaccines-13-00774-f006:**
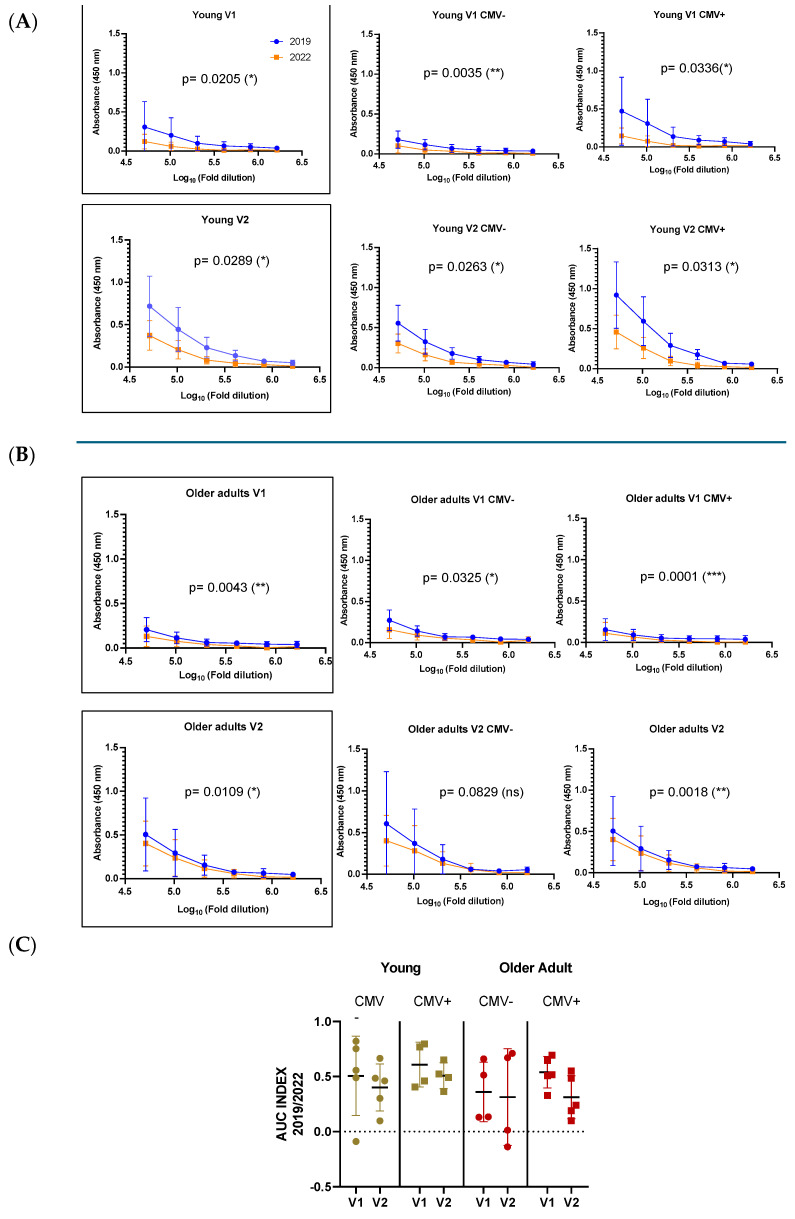
Antibody responses to A/Victoria/2570/2019 and A/Victoria/4897/2022 HA antigens in young (**A**) and older adults (**B**), stratified by CMV serostatus. Antibody titers were measured before (V1) and 28 days after vaccination (V2) using ELISA, shown as absorbance (450 nm) across serial dilutions. Blue lines indicate the 2019 antigen, orange lines the 2022 strain. The AUC index (**C**) compares responses to both strains, revealing higher 2019/2022 ratios across groups, consistent with immune imprinting and preferential recall of memory responses toward the previously encountered antigen. Data are shown as mean ± SD for curves and median ± SD for AUC. Differences were considered significant at *p* < 0.05; * (*p* < 0.05); ** (*p* < 0.01); *** (*p* < 0.001).

**Figure 7 vaccines-13-00774-f007:**
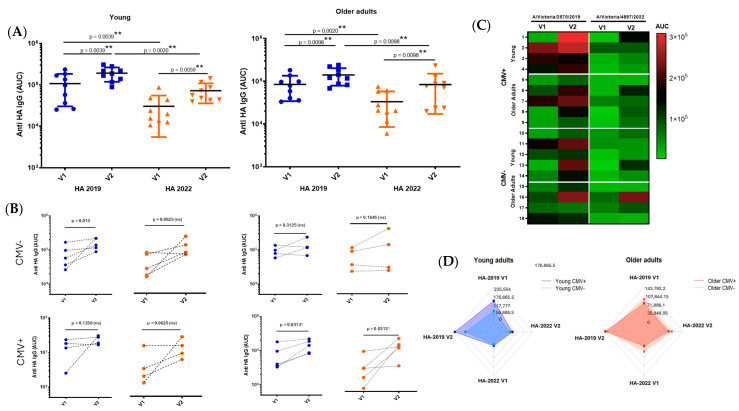
Analysis of antibody responses by age group and CMV serostatus. Antibody responses, measured as area under the curve (AUC) (**A**), were assessed before and 28 days post-vaccination, and stratified by participant age (young vs. older adults) and CMV serostatus (CMV-negative vs. CMV-positive) (**B**). The figure also includes a heat map (**C**) and a spider graph (**D**) to illustrate the response profiles across groups. (V1) Visit 1: pre-vaccination; (V2) Visit 2: 28 days post-vaccination. The results are expressed as median ± SD, and differences were considered significant at *p* < 0.05. * (*p* < 0.05); ** (*p* < 0.01).

## Data Availability

Data are available in the Appendix A. Additional data are available from the authors upon request.

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
