# Peer review of "Epitope Variation in Hemagglutinin and Antibody Responses to Successive A/Victoria A(H1N1) Strains in Young and Older Adults Following Seasonal Influenza Vaccination: A Pilot Study"

_vaccines, 2025, doi:10.3390/vaccines13070774_

Round 1
Reviewer 1 Report
Comments and Suggestions for Authors
In this manuscript, the authors employed computational approaches to predict sequence differences and potential B cell epitopes between the 2019 and 2022 HA proteins. Additionally, they compared the antibody responses elicited by the 2022 HA vaccine against both the 2019 and 2022 HA proteins. The authors suggest an imprinting effect; however, the methods are presented in a disorganized manner, making them difficult to follow. I have the following comments and recommend that the authors clarify and reorganize their data presentation in a revised version.
Major Comments:
1. In the Introduction, the authors should provide background information on how cytomegalovirus (CMV) infection may influence antibody responses following influenza vaccination.
2. In Figure 6, the meaning of “V1” and “V2” should be clearly defined. The authors should include all participant data in the same figure, with distinct groupings for younger and older individuals.
3. The source of the HA proteins used in the assays should be specified. The absorbance signal for the 2022 HA protein appears low in Figure 6. Could this be due to protein instability? The authors should discuss whether the low binding signal could be attributed to protein quality or folding issues.
Minor Comments:
1. The negative control samples should be clearly described.
2. The figure order is inconsistent and should be reorganized for clarity.
3. p-values should be formatted with a decimal point (“.”) rather than a comma (“,”).
Author Response
Reviewer 1 comments:
Comment 1: “In the Introduction, the authors should provide background information on how CMV infection may influence antibody responses following influenza vaccination.”
Response 1: We thank the reviewer for this suggestion. In the revised Introduction, we have included background information summarizing current knowledge on how latent CMV infection may affect the immune system. While its impact on influenza vaccine responses remains inconclusive—with studies reporting reduced, neutral, or even enhanced humoral responses—CMV serostatus may help explain inter-individual variability in vaccine immunogenicity.
Besides, in the Discussion section, we expanded on other potential mechanisms by which CMV might influence vaccine responses across age groups in CMV+ individuals,
Comment 2: In Figure 6, the meaning of “V1” and “V2” should be clearly defined. The authors should include all participant data in the same figure, with distinct groupings for younger and older individuals.
Response 2: Agree. The terms “V1” (Visit 1, pre-vaccination) and “V2” (Visit 2, post-vaccination) have now been defined in the legend of Figure 6 for clarity. Regarding the presentation of participant data, we chose to display each group separately in Figure 6 to facilitate visualization of individual antibody responses within each age group. However, as suggested by the reviewer, we had included a combined comparison of younger and older individuals in Figure 7, where group differences are more directly assessed.
Comment 3. The source of the HA proteins used in the assays should be specified. The absorbance signal for the 2022 HA protein appears low in Figure 6. Could this be due to protein instability? The authors should discuss whether the low binding signal could be attributed to protein quality or folding issues.
Response 3: We thank the reviewer for this pertinent observation. The sources of the HA proteins used in the assays were specified.
We agree that differences in protein quality or conformational integrity may influence the binding signal. Indeed, while our serological data consistently showed lower absorbance values for the 2022 HA antigen, this may reflect either true immunological differences or technical factors such as protein folding or stability. We have included a statement acknowledging that this is a pilot study and, although a possible antigen imprinting effect was observed, these findings require confirmation through protein quality analyses. Such analyses are critical to assess whether structural or conformational variations — independent of the primary sequence — may have influenced the observed immune response patterns, potentially accounting for the imprinting effect. This point has also been explicitly highlighted in the results and discussion sections, including the limitations of the study and future directions.
Minor Comments:
- The negative control samples should be clearly described.
Response: We appreciate the reviewer’s comment and the opportunity to clarify this point. There was a terminology imprecision in the previous version of the manuscript. What we referred to as "negative control" actually corresponds to cell control wells, which serve as background controls in the ELISA plates, rather than human serum negative controls (Sicca et al., 2022).
Obtaining truly negative human serum samples for influenza studies is inherently difficult, particularly in adult populations, due to the high likelihood of previous exposure to influenza viruses or vaccination, often undocumented. Consequently, the use of paired serum samples, collected from the same individuals before and after exposure (vaccination or natural infection), is widely recommended and considered the most reliable approach to assess specific antibody induction in this context.
This approach was implemented in our study, where serum samples from the same individuals were analyzed at two time points, allowing for the detection of changes in antibody levels over time. For the ELISA cut-off determination, we applied a method based on the mean OD of cell control wells, following the statistical approach proposed by Frey et al. (1998), which enables an objective definition of the assay background.
Additionally, to enhance clarity for the reader, we presented a simplified version of the cut-off formula, consistent with the approach used by Sicca et al. (2022). This simplification makes the calculation more accessible without compromising the scientific rigor of the method.
This part of the methods section has been rewritten to improve clarity and understanding. In addition, we have now referenced a recent publication (Sicca et al., 2022) where a similar approach for cut-off calculation based on cell control values is applied.
- The figure order is inconsistent and should be reorganized for clarity.
Response: We thank the reviewer for this observation. The figures have been reorganized and improved to enhance clarity and understanding. We have also included clear definitions of panels A, B, and C within the figure legends to facilitate interpretation. Additionally, Figure 8 has been moved to the supplementary material, as it only serves to confirm the observations presented in Figure 7 and does not provide new information.
- p-values should be formatted with a decimal point (“.”) rather than a comma (“,”).
Response: We thank the reviewer for pointing this out. The formatting of all p-values has been corrected, and decimal points are now used consistently throughout the manuscript.
All the changes were highlighted in red in the manuscript. Some new references were included
Reviewer 2 Report
Comments and Suggestions for Authors
The impact of subtle HA mutations on antibody specificity and their potential to modulate antigenic imprinting was evaluated by examining two closely related H1N1 strains from recent influenza vaccines. Linear B-cell epitopes in HA of these viruses were predicted.
The surface accessibility, hydrophilicity and structural flexibility of the predicted epitopes were analyzed. The immune response to vaccination with the A/Victoria/4897/2022 strain was studied. The immune response was compared between young and older adult participants, as well as between CMV-positive and CMV-negative participants.
Despite the fact that the vaccine contained the 2022 strain, all participants had higher antibody titers to the 2019 virus after vaccination than to the vaccine strain. The authors explain this phenomenon by antigenic imprinting between related influenza A(H1N1) strains. However, given the high similarity of the two strains and the same ratio of titers to these viruses in all participants, it seems more likely to me that in this case the lower titers to the vaccine strain can be explained by the mismatch in ELISA sensitivity for these two strains.
Note
- There is an error in the table in Figure 1. P154 is proline, not phenylalanine.
- The CMV designation must be deciphered when first used.
Author Response
Reviewer 2
Comment 1: Despite the fact that the vaccine contained the 2022 strain, all participants had higher antibody titers to the 2019 virus after vaccination than to the vaccine strain. The authors explain this phenomenon by antigenic imprinting between related influenza A(H1N1) strains. However, given the high similarity of the two strains and the same ratio of titers to these viruses in all participants, it seems more likely to me that, in this case, the lower titers to the vaccine strain can be explained by the mismatch in ELISA sensitivity for these two strains
Response: We appreciate the reviewer’s thoughtful comment. However, if the observed differences in antibody titers were primarily due to an ELISA sensitivity mismatch between the two HA antigens, we would have expected higher titers against the 2022 strain, since it matches the HA protein used in the 2022 vaccine formulation. Instead, we observed higher titers against the 2019 strain, despite both antigens being tested in the same ELISA format. This suggests that the difference in titers is unlikely to be explained solely by technical assay variability and may instead reflect biological factors, such as immune imprinting or differential recall responses. We acknowledge, however, that further experiments would be needed to formally rule out subtle differences in assay performance. In this way, the observed lower absorbance values for the 2022 HA antigen, may reflect true immunological differences or technical factors such as protein quality. We have acknowledged that, as this is a pilot study, these findings require confirmation through protein quality analyses. This has been explicitly addressed in the results, discussion, and limitations – future directions sections.
- There is an error in the table in Figure 1. P154 is proline, not phenylalanine.
Response: We thank the reviewer for carefully pointing out this error. The amino acid at position 154 is indeed proline (P), not phenylalanine (F), as incorrectly listed in the original version of Figure 1. This has now been corrected in the revised table.
- The CMV designation must be deciphered when first used.
We thank the reviewer for this observation. The full designation of CMV (cytomegalovirus) was already provided in the main text at first mention. However, we have now added the full term to the Abstract as well, to ensure clarity and consistency throughout the manuscript.
All the changes were highlighted in red in the manuscript. Some new references were included